# Enhanced Eicosapentaenoic Acid Production via Synthetic Biological Strategy in *Nannochloropsis oceanica*

**DOI:** 10.3390/md22120570

**Published:** 2024-12-19

**Authors:** Congcong Miao, Mingting Du, Hongchao Du, Tao Xu, Shan Wu, Xingwei Huang, Xitao Chen, Suxiang Lei, Yi Xin

**Affiliations:** 1School of Marine Biology and Fisheries, Hainan University, Haikou 570228, China; congcongmiao@hainanu.edu.cn (C.M.); hcdu@hainanu.edu.cn (H.D.); xtao@hainanu.edu.cn (T.X.); wushan@hainanu.edu.cn (S.W.); xvhuang@hainanu.edu.cn (X.H.); chenxitao@hainanu.edu.cn (X.C.); suxianglei@hainanu.edu.cn (S.L.); 2College of Food Science of Technology, Hainan University, Haikou 570228, China; dumingting@hainanu.edu.cn; 3State Key Laboratory of Marine Resource Utilization in South China Sea, Hainan University, Haikou 570228, China

**Keywords:** synthetic biology, marine microalgae, *Nannochloropsis*, eicosapentaenoic acid (EPA)

## Abstract

The rational dietary ratio of docosahexaenoic acid (DHA) to eicosapentaenoic acid (EPA) can exert neurotrophic and cardiotrophic effects on the human body. The marine microalga *Nannochloropsis oceanica* produces EPA yet no DHA, and thus, it is considered an ideal EPA-only model to pursue a rational DHA/EPA ratio. In this study, synthetic biological strategy was applied to improve EPA production in *N. oceanica*. Firstly, to identify promoters and terminators, fifteen genes from *N. oceanica* were isolated using a transcriptomic approach. Compared to *α-tubulin*, *NO08G03500*, *NO03G03480* and *NO22G01450* exhibited 1.2~1.3-fold increases in transcription levels. Secondly, to identify EPA-synthesizing modules, putative desaturases (NoFADs) and elongases (NoFAEs) were overexpressed by the *NO08G03500* and *NO03G03480* promoters/terminators in *N. oceanica*. Compared to the wild type (WT), *NoFAD1770* and *NoFAE0510* overexpression resulted in 47.7% and 40.6% increases in EPA yields, respectively. Thirdly, to store EPA in triacylglycerol (TAG), *NoDGAT2K* was overexpressed using the *NO22G01450* promoter/terminator, along with *NoFAD1770*–*NoFAE0510* stacking, forming transgenic line XS521. Compared to WT, TAG-EPA content increased by 154.8% in XS521. Finally, to inhibit TAG-EPA degradation, a TAG lipase-encoding gene *NoTGL1990* was knocked out in XS521, leading to a 49.2–65.3% increase in TAG-EPA content. Our work expands upon EPA-enhancing approaches through synthetic biology in microalgae and potentially crops.

## 1. Introduction

Long-chain omega-3 polyunsaturated fatty acids (PUFAs), primarily eicosapentaenoic acid (EPA) and docosahexaenoic acid (DHA), have been widely reported to be beneficial in reducing the risks of cardiovascular diseases and preventing neurodegenerative diseases [1,2]. However, the results from various studies are more or less inconsistent, leading to controversies as to the benefits of EPA and DHA [3,4]. Recently, the EPA/DHA ratio was found to be a key factor that probably affects the treatment of cardiovascular diseases [5], nonalcoholic fatty liver diseases [6], diabetes [7], and Alzheimer’s disease [8]. Moreover, DHA/EPA supplementation in a ratio of 1/1.3 has been shown to lead to EPA accumulation in cord blood during the last trimester of pregnancy [9]. Therefore, a rational dietary DHA/EPA ratio could potentially enhance cardiotrophic and neurotrophic health in humans.

For decades, DHA and EPA have primarily been sourced from marine sources like seaweed or fish [10]. Unfortunately, due to global warming, these compounds are becoming scarce for humans due to stock depletion and overfishing [11]. Moreover, due to the similar physicochemical properties of DHA and EPA, it is difficult, if not impossible to fine-tune the ratio of DHA/EPA in fish oil [12]. To tackle this problem, either EPA- or DHA-containing modules have been developed to provide pure EPA and DHA for a ratio-specific mixture [13,14]. As the initial producers of EPA and DHA, microalgae represent an attractive and competitive source, especially because of their photosynthetic system for efficient energy–biomass conversion [15]. Moreover, unlike fish oils that contain both EPA and DHA, many microalgae contain only EPA or DHA. In this context, EPA or DHA purification from microalgal extracts may be simpler than that from fish oils. On one hand, *Schizochytrium* sp. and *Aurantiochytrium* sp. have been used for industrial-scale production of DHA worldwide [16,17]. On the other hand, *Nannochloropsis* sp. is considered to be a promising EPA-producing module [18,19]. *Nannochloropsis* species are increasingly valued as models for studying microalgal lipid metabolism and as platforms for synthetic biology; however, commercial production of EPA from *Nannochloropsis*-derived sources remains unachieved. Therefore, enhancing biomass accumulation and EPA content is critical to improving EPA yields in *Nannochloropsis*.

Several approaches, such as strain improvement, fermentation process optimization, and genetic engineering, have been evaluated for increasing EPA titer and productivity [20,21,22,23]. As a derivative of genetic engineering, synthetic biology presents rational design, high efficiency, and stable modification. On one hand, studies on the EPA metabolic mechanism in *Nannochloropsis* have been initiated. A series of desaturases, elongases, isomerases, and related factors have been identified as key enzymes in EPA accumulation (Figure 1) [19,21,24,25]. On the other hand, techniques for synthetic biology, such as gene stacking, targeted gene disruption, and repression, have been developed in the *Nannochloropsis* genus [26,27]. These tools enable gene-specific, mechanistic studies and have already facilitated the engineering of improved *Nannochloropsis* strains with greater EPA production [25,28,29,30]. However, the lack of well-characterized genetic tools, including promoters and terminators, significantly hinders advancements in synthetic biology and metabolic engineering in *Nannochloropsis*. Promoters and terminators are crucial regulatory elements that control the transcription and stability of genetic constructs; however, the available resources in microalgae are limited and often species-specific. Most studies rely on a small set of endogenous promoters and terminators, which may not provide optimal expression levels or compatibility for heterologous gene expression. Furthermore, a lack of standardized characterization methods adds to the challenge, limiting the scalability of engineered pathways. Expanding the library of reliable and modular promoters and terminators for *Nannochloropsis* is essential to unlock the full potential in industrial and environmental applications [26,27]. Besides the lack of promoters and terminators resources, the current EPA yield is still far from profitable cost. Meanwhile, free EPA, which is considered to be toxic to *Nannochloropsis* cells, should be esterified for cellular storage. Thus, for the *Nannochloropsis* EPA industry, characterization of endogenous elements (e.g., promoters, terminators, enhancers, inhibitors, etc.) plus EPA-biosynthesizing modules (e.g., desaturases, elongases, isomerases, *β*-oxidation enzymes, etc.) are ever necessary.

Nitrogen starvation and high light are common stress conditions in *Nannochloropsis* cultivation. Understanding how constitutive promoters perform under such stress conditions is crucial for ensuring consistent expression of engineered pathways. This characterization is essential for developing robust genetic tools and maintaining metabolic efficiency during production processes. Therefore, in this study, to identify promoters and terminators, fifteen genes from *N. oceanica* were isolated using a transcriptomic approach. Compared to *α*-*tubulin*, *NO08G03500*, *NO03G03480*, and *NO22G01450* exhibited 1.2~1.3-fold increases in transcription level. Secondly, to identify EPA-synthesizing modules, putative desaturases (NoFADs) and elongases (NoFAEs) were overexpressed by *NO08G03500* and *NO03G03480* promoter/terminator in *N. oceanica*. Compared to the wild type (WT), *NoFAD1770* and *NoFAE0510* overexpression resulted in 47.7% and 40.6% increases in EPA yield, respectively. Thirdly, in order to store EPA in triacylglycerol (TAG), *NoDGAT2K* was overexpressed by *NO22G01450* promoter/terminator plus *NoFAD1770*–*NoFAE0510* stacking, forming the line XS521. Compared to WT, XS521 exhibited a 154.8% increase in TAG-EPA content. Finally, to inhibit TAG-EPA degradation, *NoTGL1990* was knocked out in XS521, leading to 49.2–65.3% increases in TAG-EPA content. Our work expands upon the methods for enhancing EPA via synthetic biology in microalgae and potentially higher plants.

## 2. Results

### 2.1. Transcriptome-Based Identification of Endogenous Promoters for Overexpression in N. oceanica

Promoters play a key role in the synthetic biology of microalgae. Among promoters, the constitutive promoter is considered to facilitate stable transcription and thus to maintain cellular function under both normal conditions and stresses [26]. Therefore, the constitutive promoter is an important module for EPA enhancement in this study. To identify robust constitutive promoters (RCPs), transcriptomes of *N. oceanica* were analyzed under normal conditions (N+ NL), nitrogen starvation (N− NL), and high light (N+ HL) (Section 4; Appendix A). The RNA-Seq reads were aligned to the complete genome sequence of *N. oceanica* IMET1 to produce transcriptomic data, and the resulting transcriptomic data were annotated using the IMET1 reference in the NanDeSyn website (https://nandesyn.single-cell.cn, accessed on 22 September 2024) [31].

A total of 10,333 genes were isolated from the transcriptomic data, with a mean FPKM (Fragments Per Kilobase of exon model per Million mapped fragments) value of 102. Among these genes, 63 high-transcription genes (HTGs) were identified through RPKM screening (i.e., RPKM value > 1000 under N+ NL, N− NL and N+ HL), with a mean FPKM value of 4072. By comparison with the HTG *α-tubulin* (NanDeSyn ID NO12G02410, FPKM = 1909 ± 324), the top 15 HTGs, with a mean FPKM value of 6356, were selected as the RCP candidates (Figure 2A).

To further validate the transcriptome-based identification, the transcription of the top 15 HTGs were characterized via RT-qPCR (Reverse Transcription Quantitative PCR) under N+ NL (Section 4; Appendix A). Consistent with the transcriptomic results (Figure 2A), the top seven HTGs exhibited higher transcription level than *α-tubulin*. Moreover, the transcription levels of *NO08G03500*, *NO03G03480*, and *NO22G01450* (i.e., the top three HTGs) exhibited 1.3-, 1.3-, and 1.2-fold higher values than *α-tubulin* under N+ NL, respectively (Figure 2B). On the NanDeSyn website, both NO08G03500 and NO03G03480 were annotated as light-harvesting complex (LHC) proteins, while NO22G01450 was annotated as the eukaryotic translation elongation factor EEF1A2. Thus, these three genes are considered to be highly and stably expressed, supporting cell activities under both normal and stress conditions. Therefore, the promoters of *NO08G03500* (P1), *NO03G03480* (P2), and *NO22G01450* (P3), along with their terminators (Appendix A), are considered endogenous RCPs for use in overexpression and gene stacking in *N. oceanica*.

### 2.2. Identification of EPA-Synthesizing Modules by Endogenous Overexpression of Desaturases and Elongases in N. oceanica

Fatty acid desaturases (FADs) and elongase (FAEs) are considered core enzymes in EPA biosynthesis. FAD introduce a double bond into an acyl chain by removing two hydrogen atoms, while promote fatty acid elongation, ultimately leading to the production of EPA [32]. In this study, two *FAD*s (*NoFAD0120* and *NoFAD1770*) and two *FAE*s (*NoFAE0510* and *NoFAE0440*) were isolated from the *N. oceanica* genome for subsequent EPA enhancement (i.e., PUFA FADs and long-chain FAEs). In NanDeSyn, NoFAD0120 (NO06G00120) and NoFAD1770 (NO26G01770) are annotated as “PUFA delta-5-desaturases”, while NoFAE0510 (NO29G00510) and NoFAE0440 (NO16G00440) are annotated as “long chain fatty acid elongases”. Additionally, two other FADs, NoFAD4670 (NO01G04670, delta-4 FAD) and NoFAD2680 (NO16G02680, delta-5 FAD), are also annotated in *N. oceanica*, but they are not classified as PUFA FADs [31]. Using a phylogenetic approach, the amino acid sequences of NoFADs and NoFAEs were compared with functionally validated FADs and FAEs from animals, plants, fungi, and microalgae. On one hand, microalgal FADs cluster with those animals, plants, and fungi, with NoFADs grouping alongside fungal FADs (Figure 3A). On the other hand, microalgal FAEs (including NoFAEs) form a distinct clade parallel to FAEs from animals and plants (Figure 3B). These results suggest that the aforementioned *NoFAD*s and *NoFAE*s are potential EPA-synthesizing modules in *N. oceanica*.

To probe the in vivo activities, *NoFAD0120*, *NoFAD1770*, *NoFAE0510*, and *NoFAE0440* were individually overexpressed in *N. oceanica* IMET1 (Section 4). Based on the promoter identification (Figure 2B), *NoFAD0120* and *NoFAD1770* were subcloned into plasmid pXY510, which contains the *NO08G03500* promoter and terminator, generating plasmids pXY514 and pXY515, respectively. Similarly, *NoFAE0510* and *NoFAE0440* were inserted into plasmid pXY511 (harboring *NO03G03480* promoter and terminator) to produce plasmid pXY516 and pXY517. These four plasmids (pXY514–pXY517) were then separately transformed into *N. oceanica*, resulting in the establishment of transgenic lines XS514–XS517 (Figure 4A; Section 4). Compared to the *N. oceanica* wild type (WT)*,* the *NoFAD0120*, *NoFAD1770*, *NoFAE0510*, and *NoFAE0440* transcripts exhibited 3.9~6.7-fold increases in the overexpression lines (Figure 4B). However, no significant differences (*p* < 0.01) in lipid content were observed between the overexpression lines and the WT (Figure 4C). Regarding EPA yield, while XS514 and XS517 showed no significant change (*p* < 0.01) compared to the WT, XS515 (which harbors *NoFAD1770*) and XS516 (which harbors *NoFAE0510*) exhibited increases of 47.7% and 40.6%, respectively (Figure 4D). Therefore, *NoFAD1770* and *NoFAE0510* are identified as EPA-synthesizing modules for further gene stacking in *N. oceanica*.

### 2.3. EPA Improvement by Gene Stacking in N. oceanica

Since a series of EPA-synthesizing modules (i.e., RCPs, terminators, *NoFAD*s, and *NoFAE*s) was identified through the above efforts, EPA enhancement was further investigated in *N. oceanica*. Initially, *NoFAD1770* and *NoFAE0510* were co-expressed using the *NO08G03500* and *NO03G03480* promoters and terminators in the transgenic line XS518 (Section 4; Figure 5A). Compared to *N. oceanica* WT, the transcripts of *NoFAD1770* and *NoFAE0510* exhibited 6.2- and 5.9-fold increases in XS518 (Figure 5B). Although no significant changes (*p* < 0.01) were observed in lipid content (Figure 5C), the EPA yield in XS518 increased by 45.2% (Figure 5D). Therefore, the stacking of *NoFAD1770* and *NoFAE0510* effectively improves the EPA yield in *N. oceanica*.

As previous considered, the pathway of EPA from total lipid to triacylglycerol (TAG) offers the potential to separate the source and sink, thereby providing a greater capacity for the synthesis and accumulation of EPA in microalgae [28]. Moreover, our previous studies have shown that *N. oceanica* acyl-CoA-dependent diacylglycerol acyltransferase NoDGAT2K (NanDeSyn ID NO05G02840) preferentially incorporates EPA into TAG [33]. Based on this, *NoDGAT2K* was co-expressed with *NoFAD1770* or *NoFAE0510* to generate transgenic lines XS519 (harboring *NoDGAT2K* and *NoFAD1770*) and XS520 (harboring *NoDGAT2K* and *NoFAE0510*). Additionally, *NoDGAT2K* was stacked with both *NoFAD1770* and *NoFAE0510* to produce transgenic lines XS521 (Section 4; Figure 5A).

Compared to *N. oceanica* WT, the *NoFAD1770* transcript exhibited 5.8- and 6.3-fold increases in XS519 and XS521, respectively, while no change was observed in XS520. The *NoFAE0510* transcript showed 4.4- and 5.6-fold increases in XS520 and XS521, respectively, but remained unchanged in XS519. The *NoDGAT2ZK* transcript exhibited a 6.8~9.3-fold increase in XS519–XS521 (Figure 5B). Meanwhile, total lipid content increased by 22.9% and 24.1% in XS520 and XS521, respectively, while no significant change was observed in XS519 (Figure 5C). In terms of total EPA yield, no significant change was observed in XS519 and XS520 (likely due to overlapping standard deviations), but it increased by 41.1% in XS521 (Figure 5D). Regarding TAG levels, TAG content increased by 25.2%, 20.8%, and 26.4% in XS519, XS520, and XS521, respectively (Figure 5E). Furthermore, TAG-associated EPA content showed significant increases of 89.5%, 57.3%, and 154.8% in XS519, XS520, and XS521, respectively (Figure 5F). Therefore, EPA in *N. oceanica* can be enhanced through NoFAD1770–NoFAE0510 stacking, with further accumulation in TAG via the NoDGAT2K pathway.

### 2.4. Enhancing EPA Storage by Knockout of a Triacylglycerol Lipase in N. oceanica

Although fatty acids can be stored in TAG, they can subsequently be degraded into glycerol and free fatty acids to provide energy via *β*-oxidation. The initial step of TAG breakdown is generally catalyzed by TAG-lipases (TGLs) [34]. To inhibit TAG degradation and thus enhance EPA accumulation, the TAG-lipase encoding gene *NoTGL1990* (NanDeSyn ID NO20G01990) was knocked out using CRISPR/Cas9 in *N. oceanica* (Section 4). Three CDS-indel lines, XS522-1, XS522-2, and XS522-3, were generated and characterized (Figure 6A). RT-PCR (reverse transcription PCR) results showed that *NoTGL1990* transcriptions were present in the WT but absent in XS522 lines (Appendix A), a finding that was further confirmed by RT-qPCR (Appendix A). Meanwhile, cell density remained unchanged between the *NoTGL1990*-knockout lines and *N. oceanica* WT (Figure 6B), indicating that the *NoFAD1770*–*NoFAE0510*–*NoDGAT2K* stacking, combined with *NoTGL1990* knockout, did not affect the growth of *N. oceanica*.

Regarding the total lipid level, compared to *N. oceanica* WT, the total lipid content increased by 14.3–23.9% in the *NoTGL1990*-knockout lines (Figure 6C). Additionally, the total EPA yield increased by 51.7%, 41.1%, and 38.6% in XS522-1, XS522-2, and XS522-3, respectively, reaching up to 7.8% per DW (Figure 6D). For TAG levels, TAG content improved by 41.1–55.9% in the XS522 lines (Figure 6E). Moreover, TAG-associated EPA content increased by 51.6%, 49.2%, and 65.3% in the *NoTGL1990*-knockout lines, respectively (Figure 6F). While the EPA contents in the XS522 lines was not significantly higher than in the XS521 line (Figure 5E), the enhanced TAG content and unchanged growth suggest an overall improvement in EPA yield in the XS522 lines. These findings indicate that *NoTGL1990* knockout enhances TAG content and TAG-associated EPA, which could further boost total EPA accumulation (Figure 6D). Collectively, through synthetic biology approaches, our study establishes a systematic strategy to enhance EPA production and storage, offering promising potential for utilizing *N. oceanica* as an EPA-only model to optimize the DHA/EPA ratio.

## 3. Discussion

Marine microalgae, which do not require freshwater, are typically rich in high-value compounds. Synthetic biology has gained popularity in the marine microalgae industry due to its strong relevance and high efficiency. Although genome resources for marine microalgae are developing rapidly, the lack of information regarding gene expression levels and gene functions has significantly hindered the advancement of synthetic biology in this field. Due to its high EPA content and large-scale potential, *N. oceanica* is considered an excellent platform for synthetic biology. To expand the industrial applications of *N. oceanica*, a systematic and diverse set of well-characterized expression elements is necessary for the assembly of complex biological functions or entire metabolic pathways. Prior to this study, *Nannochloropsis* promoters were predicted using densely connected convolutional neural networks [35]. However, promoter characterizations in terms of transcription, expression, and function remain limited. In this study, a total of 10,333 genes were isolated from *N. oceanica* data, and the top 15 HTGs were identified as endogenous RCPs (Figure 2A,B), which are expected to be valuable for future biosynthetic studies in *Nannochloropsis*.

High light intensity has been shown to significantly influence gene expression in microalgae, driving adaptations that enhance photosynthetic efficiency and mitigate photo-oxidative damage [36]. Genes encoding components of the photosynthetic machinery, such as LHC proteins and photosystem reaction center proteins, are upregulated to improve light absorption and energy transfer [37]. At the same time, genes involved in non-photochemical quenching mechanisms, including xanthophyll cycle-related enzymes, are highly expressed to dissipate excess light energy and prevent photo-inhibition [38]. Stress-responsive genes, such as those coding for reactive oxygen species scavenging enzymes like superoxide dismutase and catalase, are also elevated to maintain cellular redox balance under high light stress [39]. Additionally, transcription factors, including bZIP and MYB family proteins, are upregulated to coordinate light-responsive pathways [40]. These changes reflect a well-coordinated effort to enhance photosynthetic capacity, protect cellular structures, and maintain metabolic homeostasis under high light conditions. In this study, eleven HTGs, including *NO08G03500*, *NO03G03480*, *NO02G01740*, *NO12G02640*, *NO05G01620*, *NO09G00440*, *NO10G01510*, *NO21G02150*, *NO03G03690*, *NO20G00500*, and *NO16G01490*, were annotated to encode LHC proteins (Figure 2A). Understanding these HTGs is crucial for optimizing microalgae for applications in multiple biotechnological fields.

In diatoms such as *Phaeodactylum tricornutum* and *Thalassiosira pseudonana*, EPA is primarily synthetized via the omega-3 pathway. In contrast, EPA biosynthesis in *N. oceanica* occurs through the omega-6 pathway. Therefore, it is reasonable to expect that FAD overexpression may not always effective in improving EPA production in *N. oceanica*, as some FADs may not function properly with the *N. oceanica* omega-6 pathway. This “ineffective” situation has also found in diatoms. Consequently, systematic screening of functional modules is essential for advancing EPA-synthesizing biology in microalgae. In this study, two *FAD*s (*NoFAD0120* and *NoFAD1770*) and two *FAE*s (*NoFAE0510* and *NoFAE0440*) were isolated and characterized in *N. oceanica*. Compared to *N. oceanica* WT, overexpression of *NoFAD1770* and *NoFAE0510* resulted in 47.7% and 40.6% increases in EPA yields (Figure 4D). Therefore, *NoFAD1770* and *NoFAE0510* are identified as key EPA-synthesizing modules in the *N. oceanica* omega-6 pathway.

TAG is stored in lipid droplets and has the potential to accumulate at a high level in microalgae. Moreover, abiotic stress conditions, such as nitrogen starvation and high light, can triggered a drastic increase in TAG content in microalgae. Therefore, TAG-EPA storage becomes critical under stress. However, in *N. oceanica*, EPA is primarily stored in polar lipids rather than TAG. Thus, channeling EPA into TAG may represent a promising strategy to overcome the storage limitations for EPA. Prior to this study, inefficient channeling of EPA into TAG had been observed in *N. oceanica*. Co-expression of a *Chlamydomonas*-derived diacylglycerol acyltransferase gene, *CrDGTT1*, along with an elongase gene *Δ0-ELO1*, was shown to significantly increase TAG-EPA content [28]. Additionally, it was reported that knockout of the TAG lipase gene *NoTGL1* resulted in a two-fold increase in TAG content in *N. oceanica* [34]. To further enhance EPA storage in TAG, we not only overexpressed the eicosapentaenoyl-CoA-preferred diacylglycerol acyltransferase gene *NoDGAT2K* to promote TAG-EPA incorporation, but also knocked out the TAG lipase gene *NoTGL1990* to inhibit TAG-EPA degradation. As a result, TAG-associated EPA content increased by up to 65.3% in the NoTGL1990 knockout lines compared to *N. oceanica* WT (Figure 6F).

Although this study has successfully improved the accumulation of EPA in *N. oceanica*, there remains much work to be done before the *N. oceanica* EPA chassis can be fully established. To date, more than 20 desaturase/elongase coding genes have been annotated in the *N. oceanica* genome [31], but the EPA-synthesizing functions of these genes have not been systematically explored. In addition to desaturase and elongase, a range of enzymes, such as isomerases, oxidases, TAG synthases, and lipases, play crucial roles in EPA metabolism, and their functions also need to be identified. For efficient and cost-effective characterization of such a large number of gene candidates, the development of current genetic toolkits is essential. For instance, CRISPR-Cas9 technology has already been applied for gene knockout and overexpression in microalgae [41], providing a foundation for subsequent analysis of EPA metabolism and its high-yield production in *N. oceanica.* Furthermore, the development of other technologies, such as microalgal collection, EPA extraction, and EPA purification, will also be crucial for the commercialization of microalgal EPA. Collectively, in this study, we (i) identified robust promoters and terminators; (ii) characterized key FADs, FAEs, DGATs, and TGLs; and (iii) integrated these elements and functional genes to demonstrate a biosynthetic strategy for engineering EPA in *N. oceanica*. This study holds significant promise for advancing the production of PUFA in microalgae and even in crops.

## 4. Materials and Methods

### 4.1. Strains and Culture Conditions

*Escherichia coli* strain transetta (DE3) was grown in Luria–Bertani (LB) medium at 37 °C with shaking at 200 rpm. *N. oceanica* IMET1 was cultivated in modified f/2 liquid medium containing 35 g/L sea salt, 1000 mg/L NaNO_3_, 66.6 mg/L NaH_2_PO_4_·H_2_O, 3.65 mg/L FeCl_3_·6H_2_O, 4.37 mg/L Na_2_EDTA·2H_2_O, 0.0196 mg/L CuSO_4_·5H_2_O, 0.0126 mg/L Na_2_MoO_4_·2H_2_O, 0.044 mg/L ZnSO_4_·7H_2_O, 0.0109 mg/L CoCl_2_·6H_2_O, 0.036 mg/L MnCl_2_·4H_2_O, 5 µg/L VB_12_, 5 µg/L biotin, and 0.1 mg/L thiamine HCl [42]. Cells were cultivated in liquid cultures under continuous light (approximately 50 ± 5 µmol photons m^−2^ s^−1^) at 25 °C. For nitrogen starvation, *N. oceanica* cells were harvested by centrifugation (3500× *g* for 5 min) and then resuspended in N− medium (modified f/2 medium without NaNO_3_) for another 72 h. For high light induction, *N. oceanica* cells were cultivated under continuous light (approximately 200 ± 10 µmol photons m^−2^ s^−1^) for another 24 h. Cell growth was determined based on cell density using a Counterstar IC 1000.

To characterize the transgenic lines, mid-logarithmic-phase algal cells (OD_750_ of 2.6) were collected for validating successful transformants through PCR amplification and Sanger sequencing. Positive lines were then cultured for further measurements. For phenotyping, the transgenic lines, along with the WT control, were grown to an OD_750_ of 4.5 ± 0.5 over five days. Subsequently, lipid content, TAG content, and fatty acid composition were measured.

### 4.2. Identification of Promoters and Terminators in N. oceanica IMET1

To identify candidate constitutive promoters in *N. oceanica* IMET1, transcription profiles were analyzed under normal condition, nitrogen starvation, and high light induction. Total RNA from these conditions was extracted using Trizol reagents (Invitrogen, Carlsbad, CA, USA). For mRNA-Seq, the poly (A)-containing mRNA molecules were purified using Sera-mag Magnetic Oligo (dT) Beads (Thermo Scientific, Waltham, MA, USA) and fragmented into 200 to 300 bp pieces by incubation in RNA Fragmentation Reagent (Ambion, Austin, TX, USA) according to the manufacturer’s instructions. The fragmented mRNA was then purified away from the fragmentation buffer using Agencourt^®^ RNAClean beads (Beckman Coulter, Brea, CA, USA). The purified, fragmented mRNA was converted into double-stranded cDNA using the SuperScript Double-Stranded cDNA Synthesis Kit (Invitrogen) with random hexamers for priming. Strand-nonspecific transcriptome libraries were prepared using the NEBNext^®^ mRNA Library Prep Reagent Set (New England Biolabs, Ipswich, MA, USA) and sequenced for 2 × 90 bp runs (paired-end, PE) using the Illumina HiSeq2000 platform. The raw data were deposited in NCBI GEO with the reference series number GSE42508. These filtered Illumina reads were aligned to the *N. oceanica* reference genome with TopHat (version 2.0.4, allowing no more than two segment mismatches) [43]. Reads mapped to more than one location were excluded.

For each of the mRNA-Seq datasets under each experimental condition, gene expression was measured as the numbers of aligned reads to annotated genes using Cufflinks (version 2.0.4) and normalized to FPKM values. Predicted genes with expression values (FPKM) less than 10 were filtered out. To identify robust constitutive promoters, gene expression was quantified and normalized by the counterpart of *α*-tubulin (NO12G02410, FPKM = 1909 ± 324). Then, approximately 3500 genes were ranked by FPKM fold changes, and the top 35 genes were selected for further promoter characterization. The promoter and terminator sequences were identified and defined in NanDeSyn, following the previous description.

To further validate the mRNA-Seq results, RNA extracted from the same cultures used for mRNA-Seq was subjected to cDNA synthesis using the PrimeScript^®^ RT reagent Kit with gDNA Eraser (Takara, Kyoto, Japan). RT-qPCR was performed using standard methods (Roche, Basel, Switzerland) as previously described [44]. Ct values were determined for triplicate independent technical experiments, each performed on triplicate biological cultures. The primer pairs used for RT-qPCR analyses are listed in Appendix A. The sizes of amplification products ranged from 100 to 300 bp.

### 4.3. Vector Construction and N. oceanica Transformation

To construct the backbone for the *N. oceanica* IMET1 overexpression vectors, the identified promoters/terminators from *NO08G03500*, *NO03G03480*, and *NO22G01450* were amplified using genomic DNA with sequence-specific primers (Appendix A). The selected promoter and terminator regions were then inserted into pXJ450 plasmid (which harbors the zeocin-resistant gene, *BleR*) at the *Kpn*I and *Bam*HI sites, respectively. The resulting plasmids were named pXY510 (containing the promoter/terminator of *NO08G03500*), pXY511 (containing the promoter/terminator of *NO03G03480*), and pXY512 (containing the promoter/terminator of *NO22G01450*) (Figure 4A).

Furthermore, *N. oceanica* cDNA was used as the PCR templates (all primers used are listed in Appendix A). PCR products were then sequenced to obtain the full length protein-coding sequences of *NoFAD0120*, *NoFAD1770, NoFAE0510*, *NoFAE0440*, and *NoDGAT2K* (NanDeSyn ID NO06G00120, NO26G01770, NO29G00510, NO16G00440, and NO05G02840). *NoDGAT2K* was inserted into pXY512 to form pXY513. *NoFAD0120* and *NoFAD1770* were inserted into pXY510 to form pXY514 and pXY515, while *NoFAE0510* and *NoFAE0440* were inserted into pXY511 to form pXY516 and pXY517, respectively. To further construct the stacking plasmids, the expressing cassettes of P*_NO08G03500_*-*NoFAD1770*-T*_NO08G03500_* or P*_NO22G01450_*-*NoDGAT2K*-T*_NO22G01450_* were amplified and subcloned into pXY513 to form pXY518 or pXY519. Meanwhile, the expressing cassettes of P*_NO22G01450_*-*NoDGAT2K*-T*_NO22G01450_* were amplified and inserted into pXY516 or pXY518 to produce pXY520 or pXY521, respectively (Figure 5A).

To construct the CRISPR/Cas9 vectors for the *NoTGL1990* (NanDeSyn ID NO20G01990) knockout, two DNA fragments containing the hammerhead ribozyme and the *NoTGL1990* gRNA target sequence (Appendix A) were designed and annealed to form a primer dimer. The targeted sequence, ‘GACGTTTGACGCTATCCGAG’ (213–232 bp), was used with a PAM sequence of ‘AGG’. The primer dimer was then ligated to the *Bsp*QI-digested pNOC-ARS-CRISPR-v2 vector (which harbors a hygromycin-resistant gene, *HygR*) [45] to form the knockout vector pXY522. This vector expresses Cas9 and gRNA through the bidirectional *Ribi* promoter, using the *CS* and *LDSP* terminators, respectively.

Nuclear transformation of *N. oceanica* was performed for linearized overexpression vectors or the circular Cas9 vector using the high-voltage (11,000 V/cm) electroporation method. The transformants of pXY514–pXY521 were screened by zeocin (Invitrogen, 5 mg/L), while the counterpart of pXY522 was screened by hygromycin (Solarbio, 300 mg/L). Mid-logarithmic-phase algal cells (OD_750_ = 2.6) were collected for validate the successful transformants via PCR amplification (Appendix A). For *NoTGL1990*-CRISPR/Cas9 transformation in XS021, twelve PCR-positive monoclones were identified via Sanger sequencing, and ten lines were validated as positive knockout lines (Appendix A). Among these lines, three knockout types were identified: ‘GACGTTTGACGCT----GAG’ (XS022-1), ‘GACGTTTGACGC-----GAG’ (XS022-2), and ‘GACGTTTGACGCTATCCCCGAG’ (XS022-3) (Figure 6A).

### 4.4. Lipid Isolation and EPA Quantification via TLC and GC-MS

Total lipids were extracted from dried samples using chloroform/methanol (2:1 [*v*/*v*]) with a 100 mM internal control of tri13:0 TAG. The lipids were then separated on a silica TLC plate using a solvent mixture consisting of petroleum ether, ethyl ether, and acetic acid (70:30:1, by volume). To quantify the amount of TAG accumulated in *N. oceanica* WT and transgenic lines, TAG bands were scraped from the TLC plate. Fatty acid methyl esters (FAMEs) were prepared by acid-catalyzed transmethylation of the TAG bands and analyzed by GC-MS [46]. Mixed analytical standards of FAMEs and pentadecane were used as external and internal standards, respectively. The amounts of TAGs and the profiles of TAG-associated EPA were calculated based on the GC-MS results. The chemicals used as standards were purchased from Sigma (St. Louis, MO, USA).

### 4.5. Phylogenetic Analysis

The amino acid sequences of validated FADs and FAEs were aligned with the ClustalW v2.1 in MEGA4.1 [47]. The alignments were curated with GBlock [48] to remove poorly aligned positions. The curated alignment was then used to construct a phylogenetic tree using the neighbor joining (NJ) method in MEGA4.1, with the tree being tested by bootstrapping with 1000 replicates. The tree was drawn to scale, with branch lengths in the same units as the evolutionary distances used to infer the phylogenetic relationships. The evolutionary distances were computed using the Poisson correction method, with units representing the number of amino acid substitutions per site. All positions containing alignment gaps and missing data were eliminated only in pairwise sequence comparisons (using the pairwise deletion option).

### 4.6. Statistical Analysis

All experiments were conducted in triplicate, and the results are presented as the mean ± standard deviation (SD). Statistical analysis was performed using Graphpad Prism 5 (GraphPad, San Diego, CA, USA). The *p*-values were calculated using one-way analysis of variance (ANOVA). A *p*-value of less than 0.01 was considered statistically significant.

## Figures and Tables

**Figure 1 marinedrugs-22-00570-f001:**
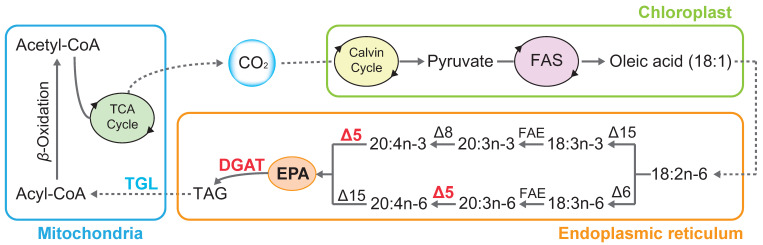
Mechanistic model of EPA metabolism in *Nannochloropsis*. Not all intermediates or reactions are displayed. Arrows indicate catalytic steps in the pathway. DGAT, diacylglycerol acyltransferase; FAS, fatty acid synthase; FAE, fatty acid elongase; TAG: triacylglycerol; TGL, TAG-lipase. The solid arrows indicate reactions occurring within the same subcellular organelle, while the dotted arrows represent reactions and transport events between different subcellular organelles. The overexpression and knockout enzymes in this study are marked by red and blue letters, respectively.

**Figure 2 marinedrugs-22-00570-f002:**
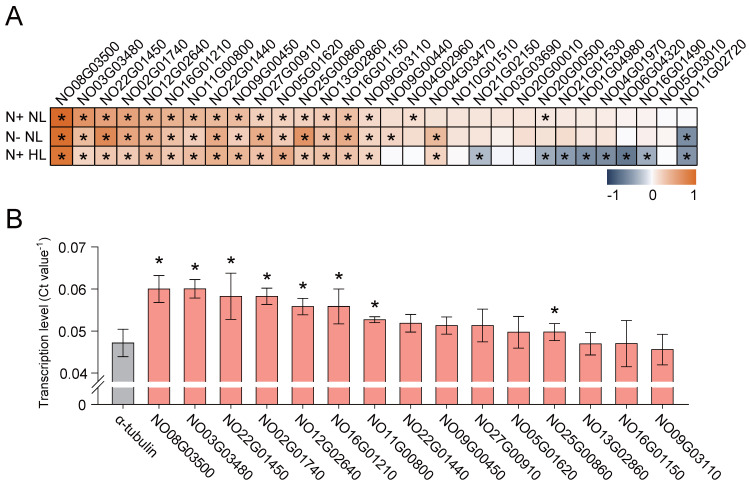
Identification of robust constitutive promoters in *N. oceanica* IMET1. (**A**) Transcriptomic response of *N. oceanica* under normal (N+ NL), nitrogen-starvation (N− NL), and high light (N+ HL) conditions. Fold change was calculated as log_2_(FPKM(Gx)/FPKM (*α-tubulin*, NO12G02410)) (FPKM = the normalized abundance of transcript; Gx = gene candidates). (**B**) RT-qPCR validation of transcript level for top 15 genes in (**A**). Gene ID was isolated from NanDeSyn (https://nandesyn.single-cell.cn, accessed on 22 September 2024). Values shown as mean ± SD (in triplicates). * significant change (*p* < 0.01) versus α-tubulin.

**Figure 3 marinedrugs-22-00570-f003:**
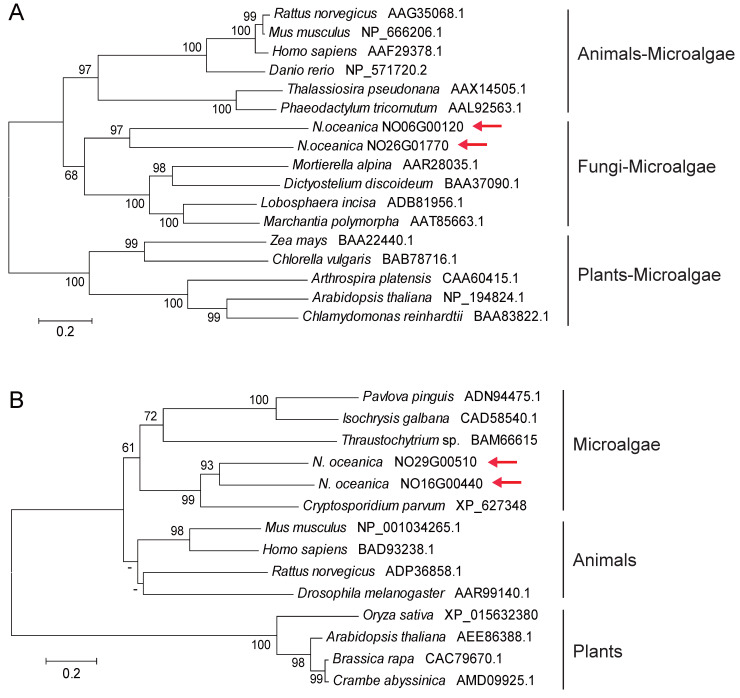
Cladograms of selected protein sequences of desaturases (**A**) and elongases (**B**) from higher plants, fungi, microalgae, and animals. Neighbor joining (NJ) was used for tree construction. Cladogram was plotted based on actual branch length. GenBank accession numbers are provided in brackets. Numbers beside the branch: bootstrap value for NJ. -, <50. Red arrow, NoFADs (**A**) or NoFAEs (**B**).

**Figure 4 marinedrugs-22-00570-f004:**
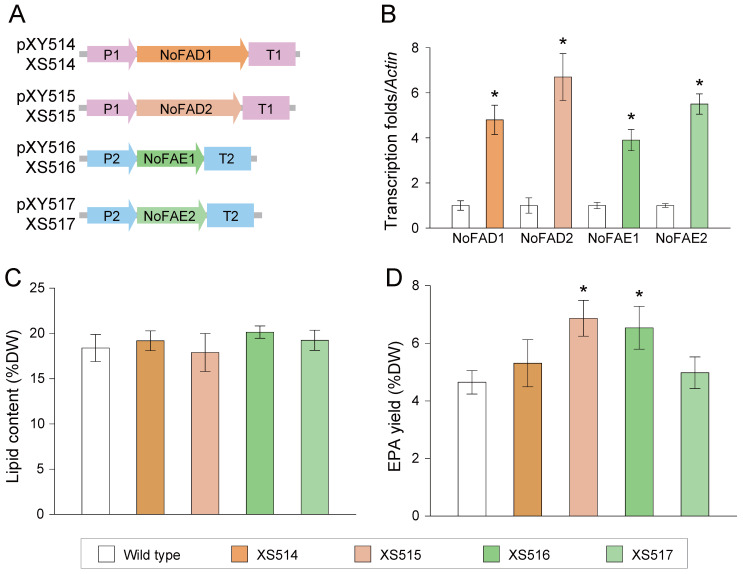
Homologous overexpression of desaturases and elongases in *N. oceanica* IMET1. (**A**) Genetic manipulation lines with overexpression of desaturases (NoFADs) or elongases (NoFAEs). (**B**) Transcript level of *NoFAD0120*, *NoFAD1770*, *NoFAE0510*, and *NoFAE0440* in *N. oceanica* wild-type and overexpression lines. (**C**,**D**) Lipid content (**C**) and EPA yield (**D**) of the *NoFAD* or *NoFAE* overexpression lines in *N. oceanica*. *NoFAD0120*, NO06G00120; *NoFAD1770*, NO26G01770; *NoFAE0510*, NO29G00510; *NoFAE0440*, NO16G00440; P1/T1, promoter/terminator from NO08G03500; P2/T2, promoter/terminator from NO03G03480. Gene ID was isolated from NanDeSyn (https://nandesyn.single-cell.cn, accessed on 22 September 2024). Values shown as mean ± SD (in triplicates). *, significant change (*p* < 0.01) versus wild type.

**Figure 5 marinedrugs-22-00570-f005:**
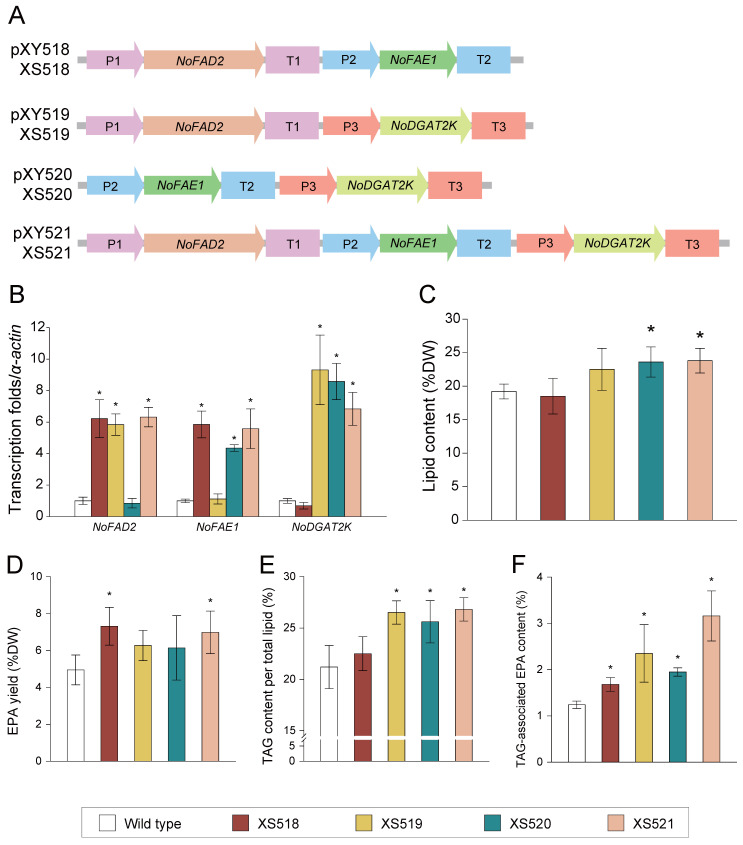
Gene stacking for EPA production in *N. oceanica* IMET1. (**A**) Gene-stacking lines harboring *NoFAD1770*-, *NoFAE0510*-, and/or *NoDGAT2K*-overexpression cassettes. (**B**) Transcript level of *NoFAD1770*, *NoFAE0510*, and *NoDGAT2K* in *N. oceanica* wild-type and gene-stacking lines, as measured by RT-qPCR. Transcription level of the above genes was normalized to that of Actin, the internal control. (**C**–**F**) Lipid content (**C**), EPA yield (**D**), TAG content (**E**), and TAG-associated EPA content (**F**) of the gene-stacking lines and wild type. *NoFAD1770*, NO26G01770; *NoFAE0510*, NO29G00510; *NoDGAT2K*, NO05G02840; P1/T1, promoter/terminator from NO08G03500; P2/T2, promoter/terminator from NO03G03480; P3/T3, promoter/terminator from NO22G01450. Gene ID was isolated from NanDeSyn (https://nandesyn.single-cell.cn, accessed on 22 September 2024). Values shown as mean ± SD (in triplicates). *, significant change (*p* < 0.01) versus wild type.

**Figure 6 marinedrugs-22-00570-f006:**
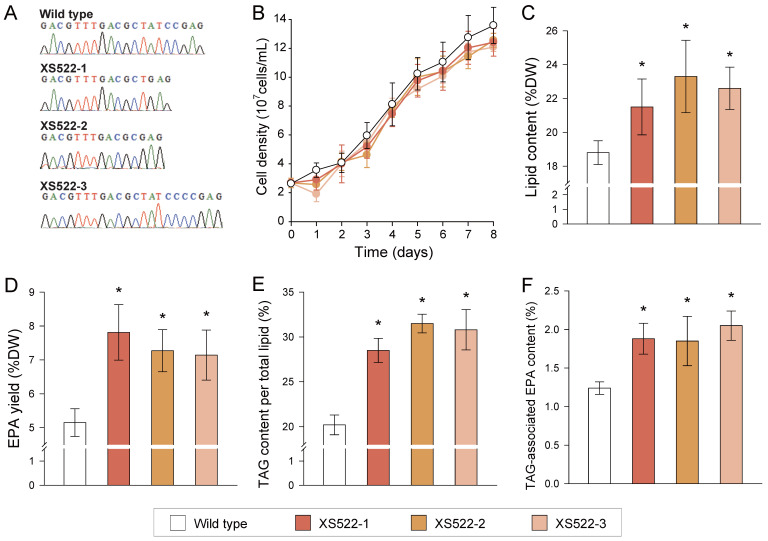
TAG-lipase knockout for EPA storage in TAG of *N. oceanica* IMET1. (**A**) Genome sequences of the editing sites in the *NoTGL1990*-knockout lines and wild type (WT). Sequences of the generated *NoTGL1990*-CRISPR/Cas9 lines confirmed the mutants as knockout lines with ‘ATCC’ deleted in XS022-1, ‘TATCC’ deleted in XS022-2, and two cytosines inserted in XS022-3. (**B**) Growth kinetics of the *NoTGL1990*-knockout lines and WT. (**C**–**F**) Comparison of the lipid content (**C**), EPA yield (**D**), TAG content (**E**), and TAG-associated EPA content (**F**) between *NoTGL1990*-knockout lines and WT. Values shown as mean ± SD (in triplicates). *, significant change (*p* < 0.01) versus wild type.

## Data Availability

The coding sequences of genes, promoters, and terminators as well as the transcriptomic data are deposited in NanDeSyn (https://nandesyn.single-cell.cn, accessed on 22 September 2024).

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
