# Peer review of "Enhanced Eicosapentaenoic Acid Production via Synthetic Biological Strategy in Nannochloropsis oceanica"

_marinedrugs, 2024, doi:10.3390/md22120570_

Round 1
Reviewer 1 Report
Comments and Suggestions for Authors
This research highlights strategies to boost EPA production and storage in Nannochloropsis oceanica via synthetic biology, with potential applications in other microalgae and crops. The study is generally well-designed and conducted. However, there are many studies where engineering strategies aimed at increasing oil content have been tested. In this study, the strategy used encompassed enhancing EPA production but also its storage in TAG as well as inhibiting TAG-EPA degradation by overexpressing an eicosapentaenoyl-CoA-preferred diacylglyceryl acyltransferase gene NoDGAT2K to reinforce TAG-EPA incorporation and knock out a TAG lipase gene NoTGL1990 to inhibit TAG-EPA degradation. The topic and the results are of interest, however, there are also some concerns to be addressed:
1.Please revise the following sentence (Line 55:) “There is a growing interest in Nannochloropsis species as models for the study of microalga lipid metabolism and as a chassis for synthetic biology, yet commercial EPA production has not achieved, Thus biomass accumulation plus EPA content should be improved to further enhance Nannochloropsis EPA yield.”
2.Please provide a separate section for the statistical tests used and complete the information regarding the tests used for the multicomparisons when using ANOVA
3.Additionally, some typos need some attention:
- Line 63 – misses a blank space before references
- Fig.1 line 116 - revise: “nitrogen-stavation”
- Fig.1 line 117 – remove double punctuation.
- Line 148 – revise: “transgeneic” check the remaining document
- Line 178 – revise “channeling”
- Line 194 – revise “indistinguished”
- Line 253 – remove the bold placed on the word “the”
- Line 259 – revise – “”synthetizd”
- Line 260 – revise “effecitve”
- Line 262 – remove de double space before FAD
- Line 278 – revise: “eicosapentaenoyl-CoA-preferrd”
- Line 283 – revise “acuumulation”
- Line 292 – revise “metaboliism”
- Line 439 - revise ref 6
Comments on the Quality of English Language
Moderate English changes required
Author Response
Dear reviewer,
Thanks so much to your suggestions, Please see attrached for the detailed response.
Best,
Yi XIN

Reviewer 2 Report
Comments and Suggestions for Authors
I found the topic and the research content of this paper is suitable for publishing in this journal after some major and minor revisions. Firstly, the introduction section did provide sufficient background for the readers, at least the author should consider include a figure describing the biosynthetic pathway of EPA (omega-3 pathway and omega-6 pathway), depict the role and the involvement of desaturases, elongases, isomerases, etc. Secondly, background information about a lack of promoters and terminators resources should be introduced here, not just briefly mentioned in the results part. Thirdly, background information about the nitrogen starvation and highlight treatment used in the study should also be properly introduced.
In the results part, the authors should tell the identity of these HTGs presented in the Figure 1a in the supplementary materials, and briefly describe the top three HTGs in the results part and include the sequences of these HTGs’ promoter and terminator in the supplementary materials.
In section 2.2, the role of the two FADs and the reason of only chosen this two FADs (and only chosen delta 5 desaturase) need to be described because in my understanding there are five desaturases involved in the whole pathway. In Figure 2a, only this two FADs were included in the phylogenetic tree, all the other FADs should be included as well.
Then in the next paragraph, line 146-159, the authors have to make a clear statement what is the relation of these genetic operations, i.e. pXY 514-pXY517, are they independent or not? Is pXY517 based on pXY516, and so on.
In Line 188, double check: is it XS520 or not?
By reading Figure 4f and Figure 5f, the increase of TAG-associated EPA in 5f is not obvious, so as the TGL knockout line has advantages or not is not obvious. Kindly ask the authors to clarify.
By reading at Figure 5a and the associated information in the results, one can not figure if these indels lead to non-functional protein or not. The authors need to show the evidences of frame-shift mutations.
In the discussion part, the author should discuss the physiological role of these HTGs in Figure 1a, and also those not very highly transcribed ones presented as blue colors in Figure 1a under the HL condition.
Some minor revisions:
Line 57: “Thus” should be “thus”.
Line 73: “β-oxidases” should be “β-oxidation enzymes”.
Line 108: delete: “the counterpart”
Line 110: delete: “the counterpart”
Line 263: correct the typo “effecitve”
Line 278: correct the name: “CrDGTT1” and “elongase gene Δ0-ELO1 ”.
Comments on the Quality of English Language
I found the topic and the research content of this paper is suitable for publishing in this journal after some major and minor revisions. Firstly, the introduction section did provide sufficient background for the readers, at least the author should consider include a figure describing the biosynthetic pathway of EPA (omega-3 pathway and omega-6 pathway), depict the role and the involvement of desaturases, elongases, isomerases, etc. Secondly, background information about a lack of promoters and terminators resources should be introduced here, not just briefly mentioned in the results part. Thirdly, background information about the nitrogen starvation and highlight treatment used in the study should also be properly introduced.
In the results part, the authors should tell the identity of these HTGs presented in the Figure 1a in the supplementary materials, and briefly describe the top three HTGs in the results part and include the sequences of these HTGs’ promoter and terminator in the supplementary materials.
In section 2.2, the role of the two FADs and the reason of only chosen this two FADs (and only chosen delta 5 desaturase) need to be described because in my understanding there are five desaturases involved in the whole pathway. In Figure 2a, only this two FADs were included in the phylogenetic tree, all the other FADs should be included as well.
Then in the next paragraph, line 146-159, the authors have to make a clear statement what is the relation of these genetic operations, i.e. pXY 514-pXY517, are they independent or not? Is pXY517 based on pXY516, and so on.
In Line 188, double check: is it XS520 or not?
By reading Figure 4f and Figure 5f, the increase of TAG-associated EPA in 5f is not obvious, so as the TGL knockout line has advantages or not is not obvious. Kindly ask the authors to clarify.
By reading at Figure 5a and the associated information in the results, one can not figure if these indels lead to non-functional protein or not. The authors need to show the evidences of frame-shift mutations.
In the discussion part, the author should discuss the physiological role of these HTGs in Figure 1a, and also those not very highly transcribed ones presented as blue colors in Figure 1a under the HL condition.
Some minor revisions:
Line 57: “Thus” should be “thus”.
Line 73: “β-oxidases” should be “β-oxidation enzymes”.
Line 108: delete: “the counterpart”
Line 110: delete: “the counterpart”
Line 263: correct the typo “effecitve”
Line 278: correct the name: “CrDGTT1” and “elongase gene Δ0-ELO1 ”.
Author Response
Dear reviewer,
Thanks so much to your suggestions, Please see attached for the detailed response.
Best,
Yi XIN

Reviewer 3 Report
Comments and Suggestions for Authors
The article ,to the best of my knowledge, is adequate. The prospect of EPA-enriched production Enhanced eicosapentaenoic acid production via synthetic biological strategy in Nannochloropsis oceanica should be carried forward. This article is crucial to the goal.
The beneficial effects in human health of lowering the ratio of docosahexaenoic acid (DHA) v.s. eicosapentaenoic acid (EPA), prospects an increase in EPA demand not sustainable by the current supply from natural marine sources, that are algae or cod liver oil. The study approaches to enhance EPA production via synthetic biology in microalgae or even crops.
The marine microalga Nannochloropsis oceanics (N. oceanica) produces EPA, and by transcriptomic approach fifteen genes were isolated alternatively (1) with greater transcription capacity of EPA production, or (2) exhibiting enhanced desaturases/ elongases enzymatic lines of EPA synthesis, or (3) storing EPA in triacylglycerol (TAG-EPA), or (4) inhibiting TAG-EPA enzymatic degradation.
What new features does it bring/what does it add that is new?
(1) the NO08G03500, NO03G03480 and NO22G01450 exhibit 1.2~1.3-folds increase in transcription; (2) NoFAD1770- and NoFAE0510-overexpression result in 47.7% and 40.6% increase in EPA; (3) TAG-EPA content in transgenic line XS521 (where NoDGAT2K is overexpressed by NO22G01450 plus NoFAD1770-NoFAE0510 stacking) increases by 154.8% compared to wild type. Finally, (4) TAG lipase encoding gene NoTGL1990 knocked out in XS521, lead to 49.2%-65.3% increase in TAG-EPA content.
Author Response

(The authors gave the same response as above.)

Round 2
Reviewer 2 Report
Comments and Suggestions for Authors
I have read through the revision. I am happy with these changes the authors made. Depends on what the other reviewers say, the editor can decide now.